

# Value of preoperative prognostic nutritional index combined with NT-proBNP in predicting acute kidney injury of congenital heart disease children

Yan Qiao, Zhenqian Lv, Xiaojun Liu, Baoguo Zhou, Haiping Wang, Gang Wang, Aiping Xie and Chenchen Cheng

Cardiovascular Surgery, Qingdao Cardiovascular Hospital, Qingdao, Shandong, China

## ABSTRACT

**Objective**. The study investigates value of preoperative prognostic nutritional index (PNI) combined with N-terminal pro-brain natriuretic peptide (NT-proBNP) in predicting postoperative acute kidney injury (AKI) in congenital heart disease (CHD) children.

**Methods**. The clinical data of 108 children with congenital heart disease were retrospectively collected. According to whether AKI occurred 48 h after operation, they were divided into AKI group ($n = 32$) and non-AKI group ($n = 76$). The clinical data, preoperative PNI and NT-proBNP levels were compared between the two groups. Multivariate logistic regression analysis was used to analyze the influencing factors of AKI, and the receiver operating characteristic (ROC) curve was drawn to evaluate the predictive value of preoperative PNI, NT-proBNP and their combination.

**Results**. Multivariate logistic regression analysis showed that Scr, PNI and NT-proBNP were independent risk factors for postoperative AKI in children with congenital heart disease ($P < 0.001$). The results of ROC curve analysis showed that the area under the curve ($AUC$) of preoperative PNI, NT-proBNP and their combination in predicting postoperative AKI in children with congenital heart disease were 0.839, 0.738 and 0.907, respectively, and the AUC of their combination was the highest.

**Conclusion**. The combined use of preoperative PNI as well as NT-proBNP holds significant value in predicting postoperative AKI in CHD children. Monitoring preoperative PNI and NT-proBNP levels may aid in clinically identifying the risk of postoperative AKI in CHD children, thereby improving their prognosis.

Corresponding author
Chenchen Cheng,
xingchen-102@163.com

## INTRODUCTION

Congenital heart disease refers to structural abnormalities in the heart formed during embryonic development, including atrial septal defect, ventricular septal defect, patent ductus arteriosus as well as tetralogy of Fallot. Congenital heart disease is a common type

of birth defect in China, with epidemiological statistics (*The Writing Committee of the Report on Cardiovascular Health and Diseases in China & Hu, 2023*) showing an incidence of approximately 8.98%, and it is on the rise, significantly impacting the safety of affected children. Currently, surgical intervention is the primary method for clinical treatment of congenital heart disease. However, postoperative acute kidney injury (AKI) is common in CHD children, with an incidence ranging from 10% to 45% (*Khuong et al., 2021*). It is closely associated with the occurrence of short-term and long-term adverse events after surgery (*Van den Eynde et al., 2021*). Therefore, early detection of AKI in CHD children after cardiac surgery while timely and effective clinical intervention are crucial for reducing mortality rates in affected children.

In the past, AKI diagnosis relied on degree of serum creatinine (Scr) elevation clinically. However, due to non-renal factors, Scr typically peaks 1 to 3 days after cardiac surgery, leading to limitations in sensitivity and other inadequacies (*Beken et al., 2021*). Research indicates (*Parker et al., 2019*) that N-terminal pro-B-type natriuretic peptide (NT-proBNP) has certain predictive value in adults AKI after cardiac surgery, but its standalone predictive value is not high. Excessive activation of systemic inflammatory response is considered a significant cause of AKI after cardiac surgery. Previous studies (*Xie et al., 2022*) have shown that albumin has multiple renal protective effects, and preoperative hypoalbuminemia may increase postoperative AKI risk. The prognostic nutritional index (PNI), calculated based on patient's plasma albumin and absolute lymphocyte count, can simultaneously reflect the patient's plasma albumin level and overall inflammatory status, demonstrating good predictive efficacy for postoperative renal complications in adult cardiac surgery (*Liu et al., 2023*).

Currently, there is no research analyzing predictive value of preoperative PNI combined with NT-proBNP in postoperative AKI in CHD children. The current study primarily aims to analyze predictive value of preoperative PNI combined with NT-proBNP in postoperative AKI in CHD children, providing a reference for clinical prevention and treatment of AKI.

## MATERIALS AND METHODS

### Subject selection

The retrospective collection of data was performed on 108 CHD children who underwent cardiac surgery between June 2020 and August 2023. Inclusion criteria demonstrated as: (1) have been diagnosed as congenital heart disease and met the indications for congenital heart disease extracorporeal circulation surgery; (2) age < 14 years; (3) complete laboratory and clinical data. Exclusion criteria were as follows: (1) Preoperative chronic kidney disease or renal dysfunction.; (2) history of renal surgery or exposure to nephrotoxic drugs; (3) emergency surgery; (4) comorbid malignancy or hematological disorders; (5) incomplete case data.

Grouping method: 108 children with congenital heart disease were divided into AKI group ($n = 32$) and non-AKI group ($n = 76$) according to whether AKI occurred 48 h after operation. AKI diagnostic criteria: According to the clinical practice guidelines for

acute kidney injury of Kidney Disease: Improving Global Outcomes (KDIGO) (*Khwaja, 2012*), AKI can be defined as one of the following conditions: (1) serum creatinine (SCr) increased by $\geq$ 0.3 mg/dl (26.5 $\mu$mol/L) within 48 h; (2) Known or assumed that renal dysfunction occurred within 7 days, SCr increased to $\geq$ 1.5 times the baseline value; (3) Urine volume < 0.5 ml/(kg h) for 6 h. This study was approved by the Ethics Committee of Qingdao Cardiovascular Hospital and abided by the ethical guidelines of the Declaration of Helsinki.

The Ethics Committee agrees to waive informed consent.

## Methods
### Collection of data
Clinical data of all eligible children were collected through electronic medical records, including: (1) surgical information, such as age, gender, body mass index (BMI), type of CHD, Risk Adjustment for Congenital Heart Surgery-1 (RACHS-1) score, preoperative left ventricular ejection fraction (LVEF), duration of surgery, aortic cross-clamp time, cardiopulmonary bypass (CPB) time, postoperative mechanical ventilation time, and intensive care unit (ICU) stay; (2) laboratory indicators: The last laboratory data of all children before surgery were collected, including Scr, 24-hour urine volume, serum albumin, total number of peripheral blood lymphocytes, and NT-proBNP. Detection method: The total number of peripheral blood lymphocytes was detected by Roche Cobas 8000 automatic biochemical analyzer (Roche, Germany). 5 ml venous blood was collected at the last time before operation, placed in a centrifuge tube without anticoagulant, placed in a 37 °C water bath for 30 min, centrifuged with a centrifugal radius of 13.5 cm, and centrifuged at 3500 r/min for 10 min. The serum was separated and stored at −80 °C. Scr level was detected by microparticle enzyme immunoassay, and the kit was purchased from Siemens (Munich, Germany). The serum albumin level of the children was detected by immunoturbidimetry. The kit was purchased from Merck Reagent Company (Germany), and the operation was carried out according to the kit instructions. The serum NT-proBNP level was detected by enzyme-linked immunosorbent assay. The kit was purchased from Merck Reagent Company (Darmstadt, Germany), and the operation was carried out according to the kit instructions.

### PNI calculation method (*Xu et al., 2022*)
The preoperative serum albumin and peripheral blood lymphocyte count of the patients were collected. The calculation formula is as follows: PNI = serum albumin (g/L) + 5 × peripheral blood lymphocyte count ($\times 10^9$/L).

## Statistical methods
SPSS software version 23.0 (IBM, New York, USA) was used to analyze the data. Normally distributed continuous variables were presented as mean ± standard deviation. Comparison between groups was conducted using independent samples $t$-test. The categorical data were presented as "$n$ (%)". Comparison between groups was conducted with the chi-square test. Multiple-factor logistic regression analysis has been used to identify the influencing factors of postoperative AKI in CHD children. Pearson correlation analysis has been

**Table 1** Comparison of surgical data between AKI and non-AKI groups.

| | AKI (n = 32) | Non-AKI (n = 76) | χ2/t | P |
|---|---|---|---|---|
| Age (years, $\overline{x} \pm s$) | 4.12 ±0.53 | 4.53 ±0.71 | 1.576 | 0.118 |
| Gender, n (%) | | | | |
| Male | 18 (56.25) | 42 (55.26) | 0.009 | 0.925 |
| Female | 14 (43.75) | 34 (44.74) | | |
| BMI (kg/m$^2$, $\overline{x} \pm s$) | 68.97 ±3.15 | 68.05 ±4.22 | | |
| Type, n (%) | | | | |
| Atrial/Ventricular septal defect | 11 (34.38) | 26 (34.21) | 0.541 | 0.910 |
| Patent ductus arteriosus | 13 (40.63) | 33 (43.42) | | |
| Tetralogy Fallot | 7 (21.88) | 13 (17.11) | | |
| Others | 1 (3.13) | 4 (5.26) | | |
| RACHS-1 grade, n (%) | | | | |
| 1 | 2 (6.25) | 8 (10.53) | 1.241 | 0.743 |
| 2 | 16 (50.00) | 39 (51.32) | | |
| 3 | 8 (25.00) | 20 (26.32) | | |
| 4 | 6 (18.75) | 9 (28.13) | | |
| LVEF (%, $\overline{x} \pm s$) | 68.60 ±6.31 | 66.97 ±6.89 | 1.150 | 0.253 |
| Surgical time (min, $\overline{x} \pm s$) | 151.48 ±11.68 | 149.26 ±11.75 | 0.898 | 0.371 |
| CPB time (min, $\overline{x} \pm s$) | 101.65 ±3.64 | 74.25 ±4.19 | 32.209 | <0.001 |
| Aortic cross-clamp time (min, $\overline{x} \pm s$) | 68.13 ±4.31 | 50.69 ±5.67 | 15.590 | <0.001 |
| Ventilation time (h, $\overline{x} \pm s$) | 11.56 ±1.82 | 4.91 ±1.33 | 21.178 | <0.001 |
| ICU stay (d, $\overline{x} \pm s$) | 9.15 ±0.46 | 4.28 ±0.85 | 30.527 | <0.001 |

**Notes.**

AKI, acute kidney injury; RACHS-1, Risk Adjustment for Congenital Heart Surgery-1; LVEF, left ventricular ejection fraction; CPB, cardiopulmonary bypass; ICU, intensive care unit.

employed to explore relationship between preoperative PNI, NT-proBNP, and Scr. The receiver operating characteristic (ROC) curve was plotted using GraphPad 8.0 software to evaluate predictive value of preoperative PNI, NT-proBNP, as wel as their combination for postoperative AKI in CHD children. $P < 0.05$ was considered statistically significant.

# RESULTS

## Comparison of surgical data between groups

Compared with the non-AKI group, the CPB time ($P < 0.001$), aortic occlusion time ($P < 0.001$), postoperative mechanical ventilation time ($P < 0.001$) and ICU hospitalization time ($P < 0.001$) in the AKI group were significantly longer, and the difference was statistically significant. No significant differences were observed in other surgical data such as gender and age between groups ($P > 0.05$, Table 1).

## Comparison of laboratory data between groups

Compared with the non-AKI group, the Scr ($P < 0.001$) and NT-proBNP ($P < 0.001$) in the AKI group were significantly higher, and the albumin ($P = 0.004$) and PNI ($P < 0.001$) were significantly lower. The difference was statistically significant. No statistically significant

**Table 2** Comparison of laboratory data between AKI and non-AKI groups.

| | AKI (n = 32) | Non-AKI (n = 76) | t | P |
|---|---|---|---|---|
| Scr (μmol/L) | 481.25 ± 80.62 | 109.47 ± 25.41 | 36.334 | <0.001 |
| 24-hour urine output (ml/kg) | 34.12 ± 13.15 | 37.88 ± 17.89 | 1.072 | 0.286 |
| Albumin (g/L) | 43.15 ± 6.97 | 46.83 ± 5.35 | 2.975 | 0.004 |
| Total lymphocyte count ($\times 10^9$/L) | 1.85 ± 0.68 | 1.90 ± 0.71 | 0.338 | 0.736 |
| PNI | 43.29 ± 6.94 | 47.02 ± 3.34 | 3.776 | <0.001 |
| NT-proBNP (pg/ml) | 1015.47 ± 162.35 | 801.36 ± 88.94 | 8.808 | <0.001 |

**Notes.**

AKI, acute kidney injury; Scr, serum creatinine; PNI, prognostic nutritional index; NT-proBNP, N-terminal pro-B-type natriuretic peptide.

differences in 24-hour urine output and total lymphocyte count between groups ($P > 0.05$, Table 2).

## Correlation analysis of preoperative PNI, NT-proBNP, and Scr in CHD children and concurrent AKI

Correlation analysis results indicated a negative correlation between preoperative PNI and Scr in CHD children and concurrent AKI ($r = -0.424$, 95% CI [$-0.655$ to $-0.121$], $P = 0.008$), as shown in Fig. 1. Furthermore, a positive correlation was observed between preoperative NT-proBNP and Scr in CHD children and concurrent AKI ($r = 0.344$, 95% CI [0.0271–0.598], $P = 0.035$), as illustrated in Fig. 2.

## Multifactorial logistic regression model analysis of postoperative AKI in CHD children

Using the variables that showed statistically significant differences in univariate analysis as independent variables and postoperative AKI in CHD children as the dependent variable (assigned as: non-AKI = 0, AKI = 1), multifactorial logistic regression model analysis revealed that Scr ($OR = 1.979$, 95% CI [1.309~2.993], $P < 0.001$), PNI ($OR = 2.017$, 95% CI [2.859~15.536], $P < 0.001$), as well as NT-proBNP ($OR = 1.698$, 95% CI [2.697~5.003], $P < 0.001$) were independent risk factors influencing postoperative AKI in CHD children ($P < 0.001$, Table 3).

## ROC curve analysis of preoperative PNI, NT-proBNP, and combined prediction for postoperative AKI in CHD children

The ROC curve showed that preoperative PNI and NT-proBNP had an area under the curve (AUC) of 0.839 (95% CI [0.751–0.927]) and 0.738 (95% CI [0.632–0.845]) in predicting postoperative AKI in CHD children. Optimal cutoff values were determined to be 44.5 and 987.3 pg/ml, respectively. The sensitivity was found to be 88.37% and 69.77%, while the specificity was 72.09% and 55.81% for PNI and NT-proBNP, respectively. When combined, the preoperative PNI and NT-proBNP showed the highest AUC of 0.907 (95% CI [0.836–0.981]), as presented in Fig. 3 and Table 4.

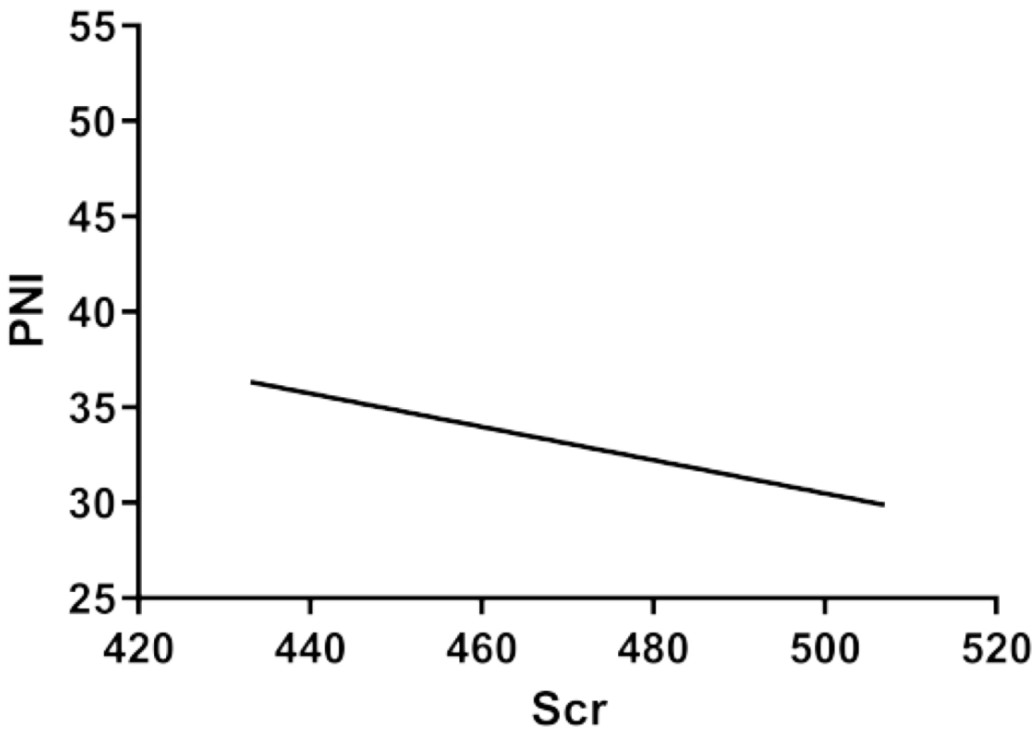

**Figure 1** Correlation between preoperative PNI and Scr in CHD children and concurrent AKI.

**Table 3** Multifactorial logistic regression model analysis of postoperative AKI in CHD children.

| Factors | β | SE | Ward χ2 | P | OR | 95%CI |
|---|---|---|---|---|---|---|
| CPB time | 0.189 | 0.102 | 3.432 | 0.082 | 1.208 | 0.989~1.475 |
| Aortic cross-clamp time | 0.100 | 0.057 | 3.068 | 0.097 | 1.105 | 0.988~1.236 |
| Ventilation time | 0.107 | 0.094 | 1.297 | 0.182 | 1.113 | 0.926~1.338 |
| ICU stay | 0.275 | 0.182 | 2.289 | 0.114 | 1.317 | 0.922~1.882 |
| Scr | 0.683 | 0.211 | 10.465 | <0.001 | 1.979 | 1.309~2.993 |
| PNI | 0.702 | 0.178 | 15.536 | <0.001 | 2.017 | 2.859~15.536 |
| NT-proBNP | 0.529 | 0.236 | 5.033 | <0.001 | 1.698 | 2.697~5.003 |

Notes.
   CPB, cardiopulmonary bypass; ICU, intensive care unit; Scr, serum creatinine; PNI, prognostic nutritional index; NT-proBNP, N-terminal pro-B-type natriuretic peptide.

## DISCUSSION

Surgical intervention is the primary clinical treatment for CHD children, effectively saving their lives. AKI is a common complication after cardiac surgery in CHD children (*Gist, Kwiatkowski & Cooper, 2018*). In the current study, a total of 108 CHD children were included, among which, 32 cases developed AKI after surgery, accounting for 29.63%, which is consistent with previous literature reports (*Xu et al., 2018*). Recent research has found that AKI not only increases the short-term mortality and incidence of chronic disease in children but also imposes a significant impact on the long-term survival rates (*Fuhrman*

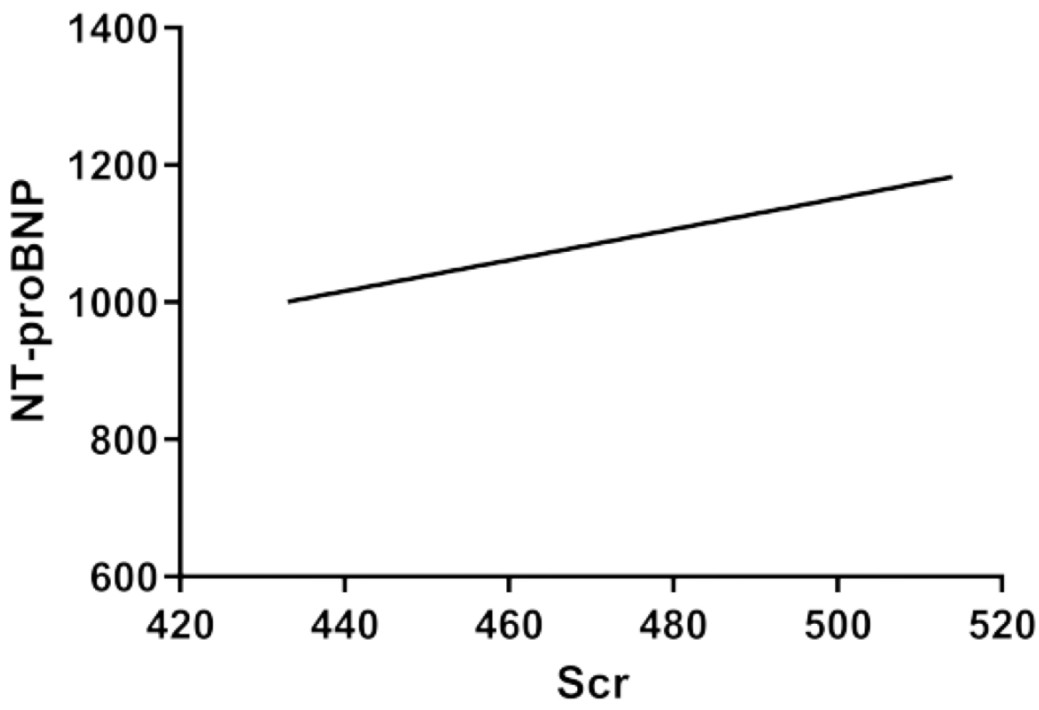

**Figure 2** Correlation between preoperative NT-proBNP and Scr in CHD children and concurrent AKI.

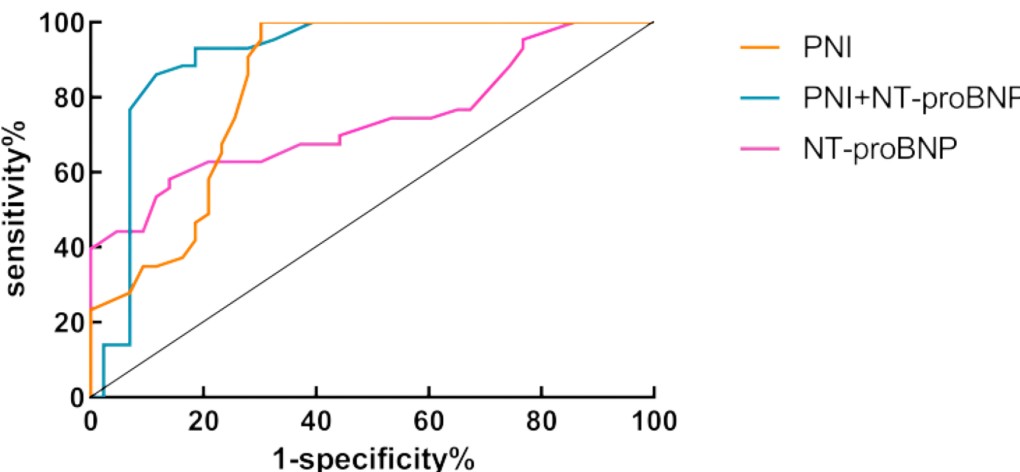

**Figure 3** ROC curve for preoperative PNI, NT-proBNP, and combined prediction for postoperative AKI in CHD children.

*et al., 2019*). A large-scale retrospective analysis study indicated that AKI is a prognostic factor for CHD children, significantly reducing their survival rates. Moreover, the impact on survival rates becomes more pronounced with AKI severity (*Nishida et al., 2019*). Therefore, predicting occurrence of AKI after surgery in CHD children is of great significance for improving their prognosis.

**Table 4   ROC curve analysis of preoperative PNI, NT-proBNP, and combined prediction for postoperative AKI in CHD children.**

|  | AUC | Optimal cutoff values | Sensitivity | Specificity | 95% CI | P |
|---|---|---|---|---|---|---|
| PNI | 0.839 | 44.5 | 88.37 | 72.09 | 0.751~0.927 | <0.001 |
| NT-proBNP | 0.738 | 987.3 pg/ml | 69.77 | 55.81 | 0.632~0.845 | <0.001 |
| Combined | 0.907 |  | 93.02 | 81.40 | 0.836~0.981 | <0.001 |

**Notes.**
PNI, prognostic nutritional index; NT-proBNP, N-terminal pro-B-type natriuretic peptide.

In this study, the regression analyses were conducted, showing that Scr was an independent risk factor for postoperative AKI in CHD children ($P < 0.001$). This finding suggests a close association between Scr levels and the occurrence of AKI in CHD children. Scr is a crucial indicator commonly used in clinical practice to assess kidney function. AKI in CHD children has decreased glomerular filtration rate, resulting in accumulation of Scr in the blood. Therefore, in cases of postoperative AKI in CHD children, Scr levels often rise, indicating an increase in serum Scr concentration, which is clinically used to diagnose AKI based on the degree of Scr elevation (*Alzahrani et al., 2022*). However, due to the kidney's reserve capacity, significant elevation of serum creatinine only occurs when renal function is severely impaired. Scr levels typically peak around 1–3 days after cardiac surgery, and it has limitations in terms of sensitivity (*Kang et al., 2018*). Therefore, it is crucial to identify new predictive indicators in this context.

Previous studies have indicated (*Kourelis et al., 2022*) that excessive activation of systemic inflammatory response is considered an important cause of AKI following cardiac surgery, and PNI is calculated according to the patient's plasma albumin as well as absolute lymphocyte count, which can reflect both the plasma albumin level and the systemic inflammatory status. Numerous studies have confirmed (*Toda & Sugimoto, 2017*; *Wang et al., 2019*) a close association between preoperative hypoalbuminemia and an increased risk of postoperative AKI in CHD children. Research has found (*Rauf et al., 2021*) that stress and malnutrition during surgery in CHD children can affect postoperative renal and immune function, and AKI can lead to protein metabolism disorders and muscle wasting, thereby affecting the nutritional status and PNI values of the children. Furthermore, postoperative AKI can impair immune function, increase the risk of infections in children, and have a negative impact on postoperative prognosis. *Sim et al. (2021)* found that the high preoperative PNI was significantly correlated with a lower rate of AKI after open liver resection in patients with hepatocellular carcinoma (95% CI [0.85–0.99], $P = 0.021$), suggesting that preoperative PNI may serve as a predictive factor for AKI, and surgical prognosis in hepatocellular carcinoma patients undergoing liver resection. *Dolapoglu et al. (2019)* indicated in their retrospective study that low preoperative PNI levels were associated with higher incidence of postoperative AKI, increased ICU admission rates, and mortality rates. In the present study, preoperative PNI was identified as the independent risk factor for postoperative AKI in CHD children ($P < 0.001$), and a negative correlation could be observed between preoperative PNI and Scr in CHD children complicated by

AKI, suggesting a close association between low preoperative PNI levels and the occurrence of postoperative AKI in CHD children.

NT-proBNP is a hormone produced by the heart and is commonly used to assess cardiac function and the severity of cardiovascular diseases. Relevant studies have indicated that postoperative AKI in CHD children can lead to renal dysfunction, decrease of glomerular filtration rate and disturbances in sodium diuresis hormones, further affecting cardiac load and influencing NT-proBNP levels (*Müller et al., 2020*). Therefore, observing changes in NT-proBNP levels can be used to evaluate the occurrence of postoperative AKI in children with congenital heart disease. However, some studies have pointed out that postoperative AKI in CHD children can interfere with metabolism and excretion of NT-proBNP, and renal impairment may lead to the accumulation of NT-proBNP in the blood, resulting in elevated levels. However, the levels can be influenced by non-renal factors, which limits the sensitivity of using NT-proBNP alone to predict postoperative AKI in CHD children (*Gong et al., 2022*).

In the current study, the value of preoperative PNI, NT-proBNP, and their combination in predicting postoperative AKI in CHD children was evaluated using ROC curves. The results showed that the AUCs for preoperative PNI and NT-proBNP in predicting postoperative AKI in CHD children were 0.839 and 0.738, respectively. Among them, the combination of preoperative PNI and NT-proBNP had the highest AUC of 0.907, indicating that the combination of preoperative PNI and NT-proBNP had a higher predictive value in for diagnosing postoperative AKI in CHD children compared to using them individually.

The shortcomings of this study are as follows: this study is a single-center retrospective study, the sample size is small, and more laboratory indicators are not included to analyze the influencing factors of AKI. In the future, multi-center and large sample size prospective studies are needed to further verify the conclusions of this study.

## CONCLUSIONS

In summary, preoperative PNI combined with NT-proBNP has a high value in predicting postoperative AKI in children with congenital heart disease. By monitoring preoperative PNI and NT-proBNP levels, it is helpful to clinically identify the risk of postoperative AKI in children with congenital heart disease. For patients with low preoperative PNI and high NT-proBNP, it is possible to consider optimizing the nutritional status in advance and strengthening the support of cardiac function, which will help clinicians improve the accuracy of postoperative AKI prediction and reduce the incidence of AKI, thereby improving the overall prognosis after surgery.

### Funding
The authors received no funding for this work.

### Competing Interests
The authors declare there are no competing interests.

## Author Contributions

- Yan Qiao conceived and designed the experiments, authored or reviewed drafts of the article, and approved the final draft.
- Zhenqian Lv analyzed the data, prepared figures and/or tables, and approved the final draft.
- Xiaojun Liu analyzed the data, prepared figures and/or tables, and approved the final draft.
- Baoguo Zhou analyzed the data, prepared figures and/or tables, and approved the final draft.
- Haiping Wang performed the experiments, prepared figures and/or tables, and approved the final draft.
- Gang Wang performed the experiments, prepared figures and/or tables, and approved the final draft.
- Aiping Xie performed the experiments, prepared figures and/or tables, and approved the final draft.
- Chenchen Cheng conceived and designed the experiments, authored or reviewed drafts of the article, and approved the final draft.

## Human Ethics

The following information was supplied relating to ethical approvals (*i.e.,* approving body and any reference numbers):

This study was approved by the Ethics Committee of Qingdao Cardiovascular Hospital and abided by the ethical guidelines of the Declaration of Helsinki.

## Data Availability

The raw data is available in the Supplementary File.

## Supplemental Information

Supplemental information for this article can be found online at http://dx.doi.org/10.7717/peerj.18085#supplemental-information.

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
