# Peer review of "Value of preoperative prognostic nutritional index combined with NT-proBNP in predicting acute kidney injury of congenital heart disease children"

_PeerJ, doi:10.7717/peerj.18085_

## Round 0.1 · original submission · Major Revisions

It is my opinion as the Academic Editor for your article - Value of Preoperative Prognostic Nutritional Index Combined with NT-proBNP in predicting Acute Kidney Injury of Congenital Heart Disease Children - that it requires some revisions as per the reviewer's comments.

Reviewer 1 ·

Basic reporting

This study mainly discussed the predictive value of preoperative PNI combined with NT-proBNP for postoperative AKI in children with coronary heart disease. The author's method is quite perfect and can get the expected results of the study well. However, there are still several issues that need to be fixed, which can improve the quality of the article

Experimental design

1.Please specify the test method of laboratory parameters, which will help you to improve the quality of your article
2.The method of grouping is not clear enough, please specify the diagnostic criteria for AKI or cite appropriate references
3.Please mark the version number and supplier of the software and instrument used in the article (city, state, country).

Validity of the findings

1.You should include references to previous studies in the discussion, which can better support the conclusions of the study.

Additional comments

1.Not all of your keywords can be found in the MeSH, please replace the keywords to ensure that the number is 3-5
2.The abstract is too long. Please reduce it to 250 words or less.
3.The limitations of retrospective studies should be explained in the discussion section. Please revise your conclusions.

Reviewer 2 ·

Basic reporting

The study investigates value of preoperative prognostic nutritional index (PNI) combined with N-terminal pro-brain natriuretic peptide (NT-proBNP) in predicting postoperative acute kidney injury (AKI) in congenital heart disease (CHD) children. The research has certain clinical significance, but there are still some issues need to be classified, and the details of the article need to be optimized.

Experimental design

In general, correlation analysis should be placed before multivariate regression analysis. Multivariate analysis was performed after clarifying significant associations between single factors.

Validity of the findings

1. Does the child have previous complications that affect renal function? Please add the relevant content in the general information.
2.The diagnostic criteria are based on the increase in the serum creatinine level and 6-hour urine output; it is best to supplement the baseline creatinine level and urine output to clarify the group assignment.
3. Why are serum creatinine, albumin, lymphocyte count, and NT-proBNP levels described separately in the Methods? Generally they can all get together through blood tests, please to reframe your inspection method.
4. In the method, the SCr and 24-hour urine output were collected postoperative, but the detection time of serum albumin, lymphocyte count and NT-proBNP was not indicated. In your statistical analysis, count them all as tests performed before surgery. Please unified your laboratory parameters testing time.
5. Please indicate in 3.1 and 3.2 specific trends between indicators that are significantly different, not just have significant differences.

Additional comments

Please define the abbreviations that appear in the table in the foot note, single see form may won't be able to understand the meaning.

Reviewer 3 ·

Basic reporting

The study fills a gap in the literature by integrating these two parameters, presenting evidence that their combined assessment provides a more reliable prediction of AKI than either measure alone. While the study presents a compelling argument, there are areas for improvement. The introduction needs to better contextualize the research within existing literature. Methodological details should be more thoroughly outlined to ensure replicability. Overall, the research offers valuable insights but requires refinement in structuring and detail to fully meet academic publication standards.

Experimental design

1)The method for calculating PNI is given, but the reason for its use is not explained.
2)The AKI diagnostic criteria should be explained in more detail with references appropriately cited. Please include the source and detailed explanation of the diagnostic criteria (e.g., citing specific guidelines or studies).

Validity of the findings

1)The statistical differences need clearer presentation, potentially including more detailed numbers and statistics. Include means, standard deviations, and exact P-values for all comparisons between groups, rather than just stating "significant differences."
2)The logistic regression results are summarized but lack specific coefficients.
3)Author should describe the protocols used for laboratory tests in more detail. Include information on the NT-proBNP assay and other laboratory methods used, such as manufacturer details, standards, and calibration procedures.

Additional comments

No comment.

---

## Round 0.2 · accepted · Accept

Dear Authors,

I have carefully reviewed the revised manuscript and am pleased to inform you that the changes you have made have satisfactorily addressed all the concerns raised by the previous reviewers. I have conducted a thorough assessment of the revision myself and am confident that the manuscript has been improved to a standard suitable for publication.

The clarity and depth of the research findings have been significantly enhanced, and the responses to the reviewers' comments demonstrate a comprehensive understanding of the feedback provided. The revisions to the methodology, results, and discussion sections have strengthened the overall quality of the paper.

In light of these improvements, I am happy to recommend this manuscript for publication in our journal. Please proceed with the final submission, ensuring that all formatting and submission guidelines are strictly followed.

Congratulations on a job well done, and I look forward to seeing your work in print.
Best regards,

Reviewer 1 ·

Basic reporting

I carefully reviewed the entire article and found that it has been well revised. I have no further comments.

Experimental design

The content of the experimental design has also been well improved, and I believe the manuscript has the level for publication.

Validity of the findings

There is nothing to modify in the survey results section.

Additional comments

No comments.

Reviewer 2 ·

Basic reporting

The article has been revised by the author, the language has become clear and smooth, the structure of the article has become professional, and the results are related to the assumptions. The references also provide sufficient on-site background.

Experimental design

After revision, the research question has become clearer, more relevant, and meaningful, demonstrating how research can fill identified knowledge gaps. The described method has become sufficiently detailed and informative to replicate.

Validity of the findings

All basic data has been provided, they are robust, statistically reasonable, and controllable. The conclusion is well presented and relevant to the original research question, limited to supporting the results.

Additional comments

no comment.

Reviewer 3 ·

Basic reporting

I think the author's revised article has been well improved in its entirety, and I have no further comments.

Experimental design

There are no other opinions on the experimental design, and the author has made revisions and improvements according to my suggestions.

Validity of the findings

The survey results and discussion section have also been well revised according to my requirements, and there are no further comments.